# Sustainable Irrigation Management of Ornamental *Cordyline Fruticosa* "Red Edge" Plants with Saline Water

**Blanca M. Plaza**, **Juan Reca**, **Juan Martínez**, **Francisco Alex** and **Maria Teresa Lao** *

Research Center on Intensive Agricultural Systems and Food Technology (CIAIMBITAL), Agri-Food Campus of International Excellence CeiA3. Ctra, Department of Agronomy, University of Almería, Sacramento s/n, La  Cañada de San Urbano, 04120 Almería, Spain
* Correspondence: mtlao@ual.es

**Abstract:** The aim of this work was to analyze the influence of the salinity of the nutrient solution on the transpiration and growth of *Cordyline fruticosa* var. "Red Edge" plants. A specific irrigation management model was calibrated with the experimental data.  An experiment was performed with four treatments. These treatments consisted of the application of four nutrient solutions with different electrical conductivity ($EC_w$) levels ranging from 1.5 dS m$^{-1}$ (control treatment) to 4.5 dS m$^{-1}$. The results showed that day-time transpiration decreases when salt concentration in the nutrient solution increases. The transpiration of the plant in the control treatment was modelled by applying a combination method while the effect of the salinity of the nutrient solution was modelled by deriving a saline stress coefficient from the experimental data. The results showed that significant reductions in plant transpiration were observed for increasing values of $EC_w$. The crop development and yield were also affected by the increasing salinity of the nutrient solution. A relationship between the $EC_w$ and the relative crop yield was derived.

**Keywords:** crop growth model; irrigation management; salinity; stress coefficient; transpiration model

## 1. Introduction

The growing of flowers and ornamental plants is an important production sector in many parts of the world. The total area devoted to grow ornamental plants in Spain is approximately 5636 ha and its economic importance is very high [1].

The efficient management of the water and nutrients is essential not only to achieve higher yields and economic returns, but also to ensure the environmental sustainability of this important production sector [2].  The improvement of irrigation water efficiency is of paramount importance, especially in semiarid regions, like some areas located in the Mediterranean Basin in Spain, especially in the greenhouse production district, and its importance will increase in the future due to climate change. Water resources are scarce and the withdrawal of water from aquifers causes severe overexploitation. An increasing salinization process [3,4] and non-point source pollution of the groundwater due to nitrate is taking place because of the irrigation return flows [5]. This situation is common in many regions since, according to the Food and Agriculture Organization of the United Nations (FAO) [6], over 6% of the world's land, which accounts for more than 800 million ha, is affected by either salinity or sodicity.

The development of efficient and sustainable methods to manage salinity in intensive semiarid irrigation systems is an active field of research.  Among these methods, the use of closed soilless cropping systems with recirculation of leachates or cascade cropping systems [7] have been proposed by

several researchers as an efficient way to improve the water use efficiency (WUE) in intensive semiarid irrigation systems affected by severe salinization [8]. Other scientists have proposed intercropping halophytic species, able to accumulate salts in their tissues, to mitigate salinity stress in a watermelon crop [9].

*Cordyline fruticosa* var. "Red Edge" belongs to the *Agavaceae* family. This family are integrated for plants that have a metabolism specialized in saving water, namely Crassulacean Acid Metabolism (CAM metabolism). This plant is well adapted to the Mediterranean climate and tolerates a certain level of salinity [10–12]. Nevertheless, currently, there are no specific studies on water consumption, especially in saline conditions that make it possible to automate irrigation.

Water outputs from the substrate in pot growing systems consist of evaporation, leaching, and plant uptake [13]. The amount of water evaporated and the evaporation rate in container crops are conditioned by several atmospheric factors, such as solar radiation, air humidity, air temperature, and wind speed, as well as the surface of evaporation [14]. A leaching fraction value ranging from 10% to 30% is commonly used depending on the season, plant development, and water quality [13].

Plant water uptake is mainly due to the transpiration of the crop. The transpiration rate is also controlled by physical factors that determine the climatic demand (solar radiation, vapor pressure deficit) and by several physiological (stomatal conductance, growth rate) and morphological factors (leaf area, leaf angle). In soilless culture, the root volume is small and the amount of available water for the plant is low, so mistakes in water application cannot be buffered by the system. That is why improving irrigation water use in container crops is a very important issue [15].

Among the methods proposed by the FAO [16] to determine reference evapotranspiration, the most commonly used methods to calculate the transpiration of crops in greenhouse conditions are the Penman–Monteith method and pan evaporation methods. The approach proposed by Penman–Monteith has been applied to greenhouse conditions [17–19]. However, most of their work was devoted to the study of vegetable crops. Moreover, the environmental conditions under which most of these studies were conducted is considered to be mild compared with those that exist in greenhouses in warmer areas. For instance, Bakker [18] studied the effect of humidity on the leaf conductance of four glasshouse vegetable crops, but his experimental ranges for temperature, photosynthetically active radiation (PAR), and vapor pressure deficit (*VPD*) were far below those existing in Mediterranean greenhouses [20].

In Mediterranean conditions, Fernández et al. [21] found that these methods estimated with similar precision the reference evapotranspiration ($ET_0$) under plastic greenhouses. They also concluded that solar radiation was the main factor affecting $ET_0$, so they developed and proposed a radiation method, locally calibrated, that was shown to perform correctly under these specific conditions. In addition to the Penman–Monteith equation, another simplified combination method widely accepted to estimate the hourly transpiration rate under greenhouse climatic conditions is that proposed by Baille et al. [22]. This method was successfully applied to roses [23] and ornamentals [19].

However, none of these models consider the effect of the osmotic potential of the nutritive solution. The water for irrigation purposes in Mediterranean countries usually contains high levels of chloride, sodium, sulfate, calcium, and other ions [24] and there is a risk of salt accumulation in the soil, especially for long growing cycles and warm growing seasons (spring and summer seasons) and when growers are forced to use saline water [25].

The decrease of the substrate water potential due to the salinity of the nutrient solution is expected to cause reductions in the actual transpiration ($T_a$) of the crop [26] and this reduction in transpiration is also expected to reduce crop yields [27–29]. Zhang et al. [30] analyzed the saline stress in two shrub species in China under different levels of salinity. These authors found that the photosynthetic rate, transpiration, and WUE decreased significantly with increasing salt concentrations. Hossain and Nonami [31] observed that salt stress can also affect not only the fruit growth rate but also cuticle permeability and induction of blossom-end rot in a study performed on hydroponically grown tomato fruit.

There is a need to establish how root water uptake should be calculated under saline conditions, and to test calculated uptake against experimental data recorded under documented site conditions. Maas and Hoffman [27] proposed a linear model to account for salinity stress. A threshold electrical conductivity value, $EC_T$ (dS m$^{-1}$), was defined, below which no salinity stress is assumed. At salinity levels above $EC_T$, the root water uptake declines at a constant rate. Salinity stress can be described by means of a stress coefficient ($K_s$). Essentially, $K_s$ is a modifier of its target model parameter, and varies in value from one (no stress) to zero (full stress) [32]. In addition to the linear shaped curve proposed by Maas and Hoffman [27], other salinity stress curves with different shapes have been proposed in the literature. The model Aquacrop [33–35] recommended by the FAO proposes three different shapes for the salinity curves, i.e., linear, convex, and logistic. The type of stress curve depends on the behavior of each crop and its parameters must be set by experimental calibration and should be based on knowledge of the crop's salinity resistance or tolerance. No previous general recommendations are made regarding the most appropriate shape and parameter values for different types of crops and cropping systems. For this reason, it is very important to experimentally analyze and calibrate the most fitted salinity stress curve for a potted plant like the selected *Cordyline fruticosa* var. "Red Edge".

Other more complex crop simulation models, such as ECOSYS [36], SWAP [37], or Aquacrop [33], have not been tested and calibrated to quantify the actual transpiration of potted plants as a function of the salinity of the nutrient solution in greenhouse conditions.

In this work, the influence of the salinity of the nutrient solution on the transpiration of the plant *Cordyline fruticose* and on its irrigation needs and yield was experimentally analyzed and mathematically modeled with the aim of deriving practical recommendations about the irrigation management of this kind of potted plants.

This paper aimed to analyze experimentally the influence of the osmotic potential of the fertigation solution on the transpiration reduction of a specific potted *Cordyline fruticosa* var. "Red Edge" plants and the relationship between the osmotic potential and the induced transpiration reduction of the crop, and the crop growth and yield. This study was carried out under Mediterranean greenhouse conditions. A specific model was calibrated with the experimental data. This model can be applied in practice by farmers to improve the irrigation efficiency of their potted plants crops.

## 2. Materials and Methods

### 2.1. Plant Growth Environment

The trial was carried out in a multitunnel greenhouse type INSOLE (Buried Solar Greenhouse), described by Lao et al. [38], with zenithal ventilation and a shading screen, located in the province of Almería (Spain) in the Mediterranean area.

### 2.2. Crop Conditions

In total, 64 young plants of a 12 cm height in 10 cm diameter pots with *Cordyline fruticosa* "Red Edge" plants (one plant per pot) were used. The crop density was 30 plants m$^{-2}$. The substrate used was a mixture of peat and perlite (80:20 *v/v*), a combination that is commonly used by farmers in the area. Four pots were placed on a tray. The trays were covered with a plastic film to avoid evaporation losses from the pots, so that uptake was equal to transpiration and with a black/white plastic sheet to avoid algae proliferation and excessive heat in the root media. The nutritive solutions were applied manually, the frequency depending on plant requirements. The trial was conducted from 24 May to 27 June (4 weeks). The crop was always in a vegetative phenological state.

### 2.3. Treatments

The basic fertigation solution (control treatment, $T_0$) had a pH of 8.0, with 3.0 mmol L$^{-1}$ of N-NO$_3^-$, 1.4 mmol L$^{-1}$ of H$_2$PO$_4^-$, 2.0 mmol L$^{-1}$ SO$_4^{2-}$, and 9.9 mmol L$^{-1}$ of K$^+$. Salinity was imposed by the addition of salt (NaCl) to $T_0$ in order to obtain EC$_w$ values of 1.5, 2.5, 3.5, and 4.5 dS m$^{-1}$. Table 1 shows

the $Na^+$ and $Cl^-$ concentrations of the nutrient solutions used in this trial and the osmotic potential ($\psi^0$) estimated by the equation, $\psi^0$ (MPa) = $-0.036$ $EC$ (dS m$^{-1}$), proposed by the United States Soil Salinity Laboratory (USSL) [39].

**Table 1.** Electrical Conductivity (dS m$^{-1}$), osmotic potential ($\psi^0$) (KPa), and $Na^+$ and $Cl^-$ concentrations (mmol L$^{-1}$) of the nutrient solutions.

|       | *EC.* | $\psi^0$ | $Na^+$ | $Cl^-$ |
|-------|-------|----------|--------|--------|
| $T_0$ | 1.5   | $-54$    | 14.3   | 11.6   |
| $T_1$ | 2.5   | $-90$    | 22.2   | 22.4   |
| $T_2$ | 3.5   | $-126$   | 32.7   | 31.8   |
| $T_3$ | 4.5   | $-162$   | 38.2   | 37.4   |

## 2.4. Experimental Design and Statistical Analysis

The experimental design consisted of 4 treatments: 1.5 dS m$^{-1}$ ($T_0$, control), 2.5 dS m$^{-1}$ ($T_1$), 3.5 dS m$^{-1}$ ($T_2$), and 4.5 dS m$^{-1}$ ($T_3$), the usual range of salinity in the Mediterranean area where *C. fruticosa* presents an adequate level of salinity tolerance. The plants were placed in trays with 4 pots per tray (each replicate consisted of one tray, and there were 4 replicates per treatment with a total of 16 plants per treatment). The trays were arranged in a randomized design. A line of plants was placed surrounding the other plants that constituted the trial in order to avoid the borderline effect. Analysis of data was made using the software packages, Excel 7.0 and Statgraphics Centurion.

## 2.5. Sampling and Data Recording

(a)　Climatic conditions: For the whole experiment, the air and substrate temperature, radiation, and *VPD* were monitored on an hourly basis with an Onset HOBO LCD data logger model H8 RH/Temp/Light/External H08-004-02, with a probe TMC6-HA (substrate temperature) and a model PYR Apogee pyranometer sensor.

(b)　Night and day water uptake: For the whole experiment, night and day plant weights were measured in order to determine the transpiration of each replicate. A scale Dicsa MonoBloc incide PB602-S was used. The plants' weekly water uptake was measured at the end of the four weeks of experiment with the aim of analyzing the evolution of the transpiration along the experiment. Night and day water uptakes were measured separately in order to assess how the salinity of the nutrient solution affects the transpiration of the plant during dark and light periods. From the measured data, average daily water uptakes for each week were calculated.

(c)　Transpiration dynamics: The weight of each plant was measured each hour for a period of 25 h in order to determine the hourly transpiration.

(d)　Leaf area index (*LAI*): it was measured via digitalization of the leaves with a scanner HP Deskjet 990CXI and the subsequent image treatment was done with the IDRISI software package for Windows version 2007. LAI was estimated from the leaf area measurement considering the plant density.

(e)　Aerial dry weight: Leaves and stems were jointly dried in a NüveE FN500 oven at 60 °C for 48 h at the end of the experiment to determine the dry weight with a scale Dicsa MonoBloc incide PB602-S.

## 2.6. Modeling Actual Transpiration

The potential transpiration of the crop under no saline conditions was modeled using the equation proposed by Baille et al. [19,22], which has been widely used for soilless crops under greenhouse conditions:

$$T_c = A \times f_1(LAI) \times R + B \times f_2(LAI) \times VPD, \tag{1}$$

where $T_c$ (g m$^{-2}$ h$^{-1}$) is the crop transpiration under no stressed conditions (T$_0$ treatment), $R$ is the canopy solar radiation received (gm$^{-2}$ h$^{-1}$), and *VPD* is the vapor pressure deficit (kPa) inside the greenhouse, with $f_1$ and $f_2$ functions of the *LAI* (m$^2$ m$^{-2}$). Function $f_1$ is an exponential regression function that represents the classic relationship of radiation interception by the canopy (dimensionless) and is a function of the extinction coefficient of radiation that passes through the canopy ($k$), as shown in Equation (2):

$$f_1 = 1 - e^{-(k \times LAI)}. \tag{2}$$

Function $f_2$ is equivalent to the *LAI*, with *LAI* considered as a multiplicative factor of the advective component in the Penman–Monteith equation. *A* and *B* are crop related coefficients, which must be determined experimentally for every crop, with *A* being a non-dimensional term and *B* expressed in g m$^{-2}$ h$^{-1}$ k Pa$^{-1}$. Parameters *A* and *B* were adjusted with the experimental data obtained for non-stressed saline conditions (treatment T$_0$).

### 2.7. Saline Stress Curve

The effect of salinity was modelled considering a salinity stress coefficient:

$$T_a = k_s \times T_c, \tag{3}$$

where $T_a$ is the actual transpiration (g m$^{-2}$ h$^{-1}$) and $K_s$ is the salinity stress coefficient.

The electrical conductivity of the soil solution ($EC_s$) is an appropriate parameter to quantify the salinity conditions in the substrate [38]. However, in this experiment, plants were irrigated very frequently and with a high leaching fraction and under these conditions, $EC_s$ and $EC_w$ were approximately the same, so we preferred to relate the salinity stress coefficient directly to $EC_w$. $K_s$ values for every level of $EC_w$ were experimentally obtained by applying a linear regression between the $T_a$ values of plants measured for the corresponding treatment and the potential crop transpiration under no saline conditions ($T_o$).

The experimentally obtained $K_s$ values for every level of $EC_w$ were then adjusted to the saline stress curves proposed in the literature [35,36] and to a new shape proposed in this work with the aim of selecting the best suited to describe the behavior of the Plant

### 2.8. Modeling the Relationship between Salinity and Crop Yield

Plants under salinity generate a complex response that differs from one family to another. For that, in this work, instead of analyzing directly the effect of the salinity of the crop yield as other researchers have proposed [9,27,40,41], we preferred to separately consider the two processes involved. First, we studied the experimentally observed relationship between the salinity of the nutrient solution and the transpiration deficit, and then we analyzed the effect of the transpiration deficit on the relative crop yield, expressed as experimental aerial dry matter, and following the approach described in other previous works [15,42].

## 3. Results

### 3.1. Environment Conditions

Table 2 shows the climatic conditions and root temperature during the trial. It is remarkable that the average air and root temperatures were similar. In this trial, *VPD* ranged from −0.60 to −3.56 kPa, with an average value of −1.80 kPa.

**Table 2.** Climatic conditions and root temperature during the trial.

| | Climatic Conditions | | | Root Conditions |
|---|---|---|---|---|
| | **Global Radiation (W m$^{-2}$)** | **Temperature (°C)** | ***VPD* (kPa)** | **Temperature (°C)** |
| Average of max. | 550 | 35.46 | −3.56 | 34.31 |
| Average of min. | 0 | 19.46 | −0.60 | 20.38 |
| Average of aver. | 170 | 26.76 | −1.80 | 26.75 |

### 3.2. Water Uptake

The higher night water uptake in the saline treatments did not make up for the lower daily water uptake under salinity. The results are shown in Figure 1.

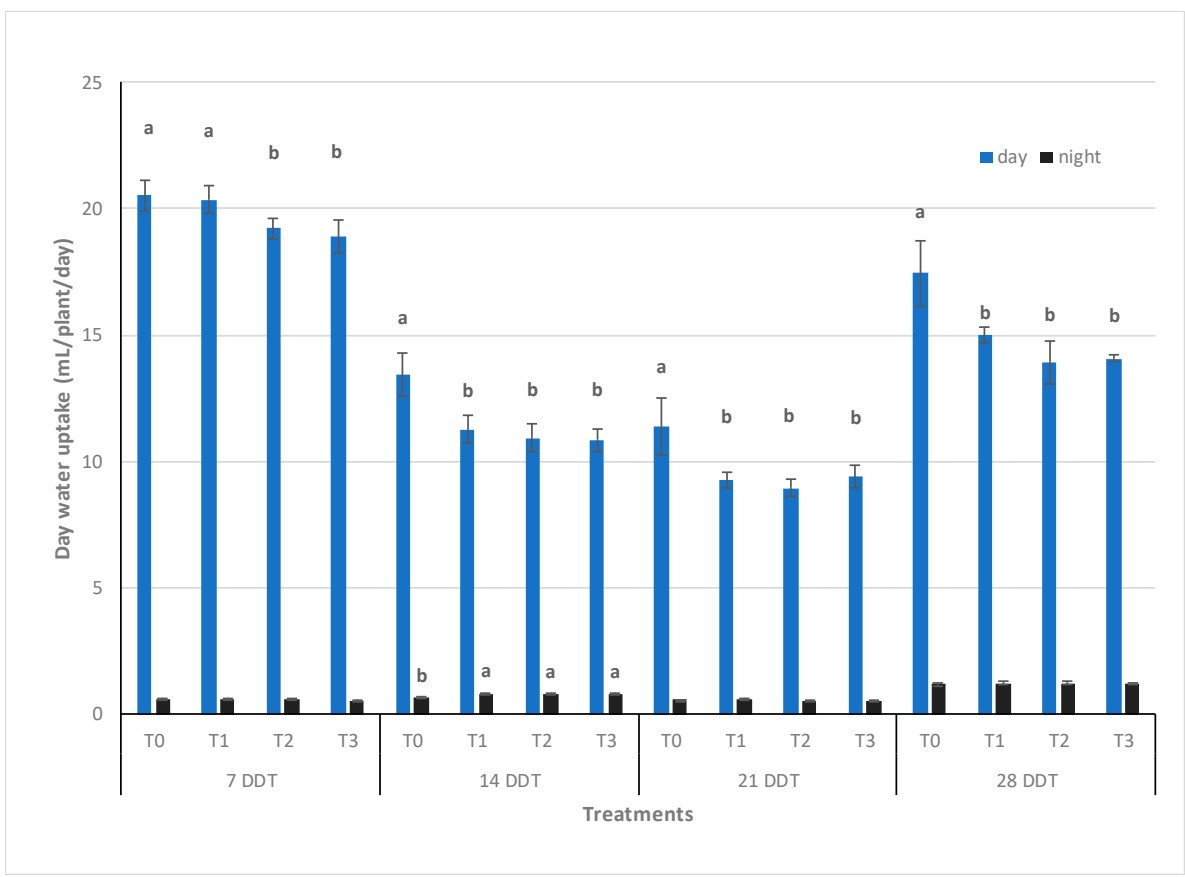

**Figure 1.** Average night and day water uptake per plant for every week and treatment.

The average daily water uptake per plant during the trial was 18.8 (100%), 16.8 (89%), 16.0 (85%), and 15.6 (83%) mL day$^{-1}$ for treatments $T_0$, $T_1$, $T_2$, and $T_3$, respectively. Control plants ($T_0$) had a significantly higher transpiration than saline treatments ($p < 0.05$). $T_1$ showed differences with $T_2$ and $T_3$ except in the fourth week. However, there were no significant differences ($p > 0.05$) between $T_2$ and $T_3$. On the other hand, daily water uptake accounted for 97% of the total water uptake. This water uptake was significantly lower in $T_0$ than in the saline treatments during the second week.

### 3.3. Potential Crop Transpiration (TC)

Figure 2 shows the dynamic relationships between transpiration and the main climate parameters in the combination methods (radiation and *VPD*) for the control treatment, $T_0$:

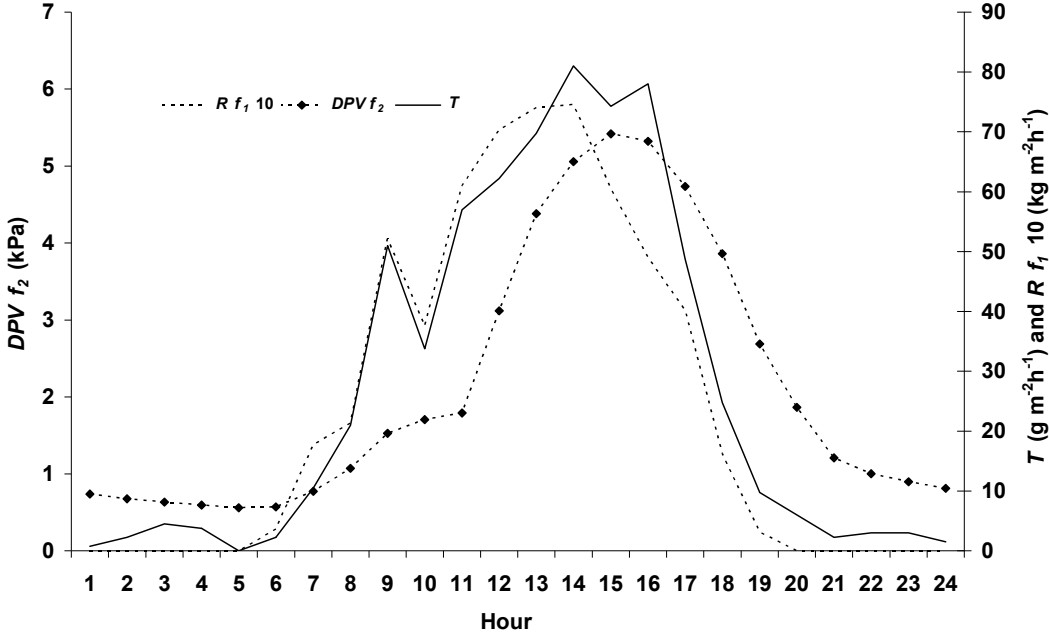

**Figure 2.** Experimental and estimated transpiration related to addends modified by Baille et al. [19,22] of the Penman–Monteith equation (temperature $* f_1 * 10$ and DPV $* f_2$).

The experimental data (n = 64) for the control treatment were fitted to the transpiration equation proposed by Baille et al. [19,22] and the resulting coefficient of determination ($R^2$) was equal to 0.98 ($p < 0.05$). The adjusted equation proposed to calculate transpiration of the *C. fruticosa* under non saline conditions and in Mediterranean greenhouse climate conditions is the following:

$$T_c = 99.69 \times f_1(LAI) \times R + 4.40 \times f_2(LAI) \times VPD. \tag{4}$$

### 3.4. Transpiration Dynamics and Salinity

Figure 3 shows the transpiration dynamics measured for each treatment in an hourly basis. It is interesting to highlight that there was a lack of response of plants $T_3$ and $T_4$ to solar radiation variations between 8 and 12 h (dotted line) (Figure 3).

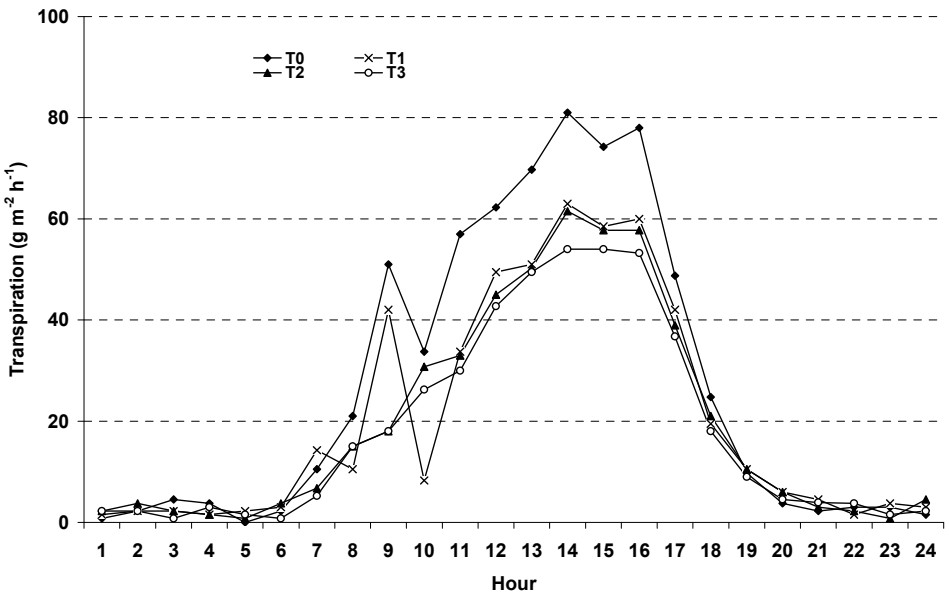

**Figure 3.** Experimental daily transpiration pattern of treatments essayed.

### 3.5. Relationship between Salinity and Yield

With the aim of modelling the actual transpiration of the plant under saline conditions, a linear regression fit wasperformed between the actual transpiration ($T_a$) for each treatment and the measured crop transpiration ($T_c$) under no saline conditions (treatment $T_0$). Linear relationships between both variables for every salinity level were found since the experimental data fitted well to the straight lines intersecting the origin with coefficients of determination equal to 0.98 and *p*-values lower than 0.05 in all cases. Figure 4 shows the result of the linear regression fits for treatments $T_1$ (a), $T_2$ (b), and $T_3$ (c). The slope of the linear regression represents the saline stress coefficient, Ks, value obtained for each treatment. Values of 0.79, 0.74, and 0.69 were obtained for treatments $T_1$ (2.5 dS m$^{-1}$), $T_2$ (3.5 dS m$^{-1}$), and $T_3$ (4.5 dS m$^{-1}$), respectively.

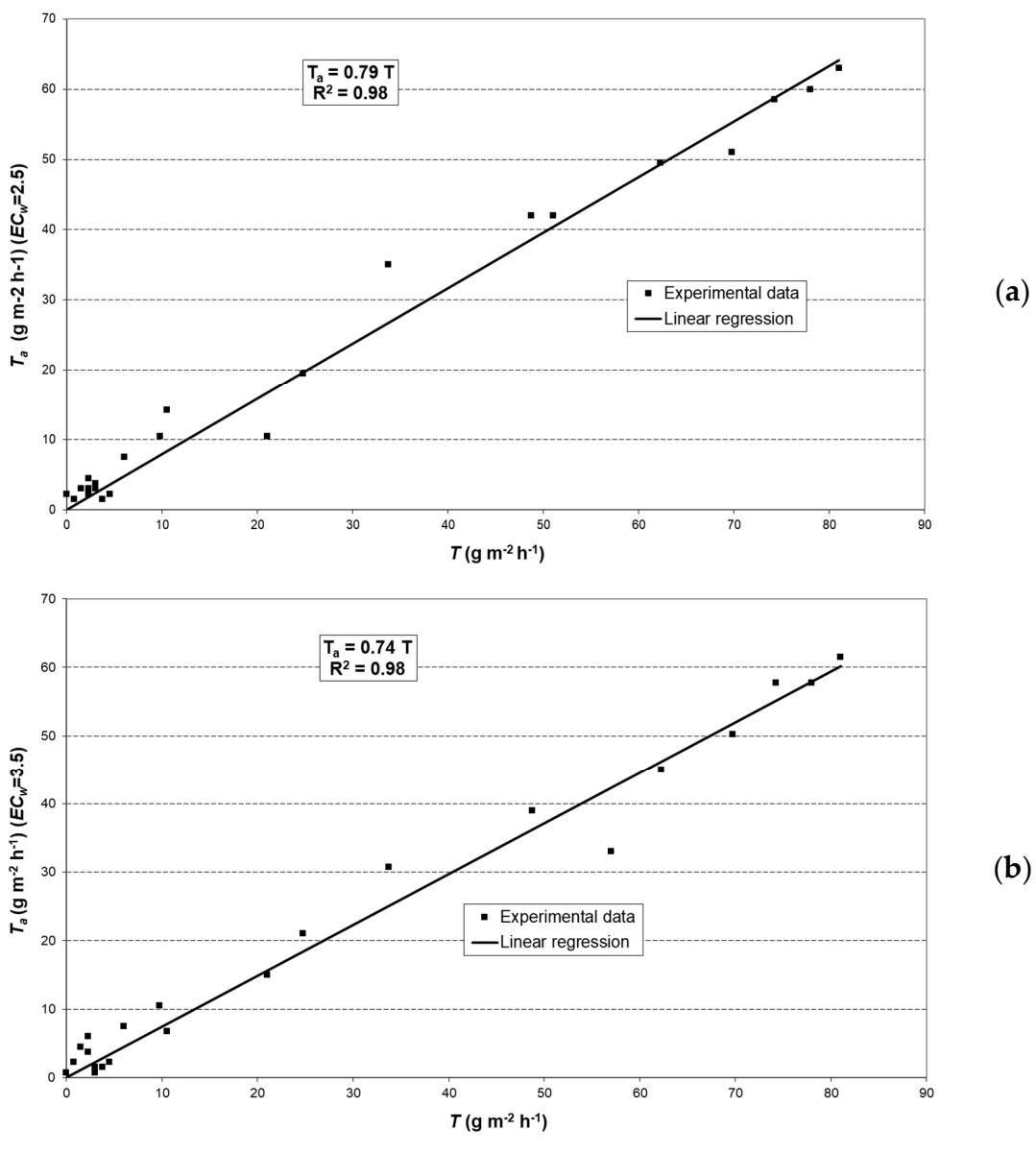

**Figure 4.** *Cont.*

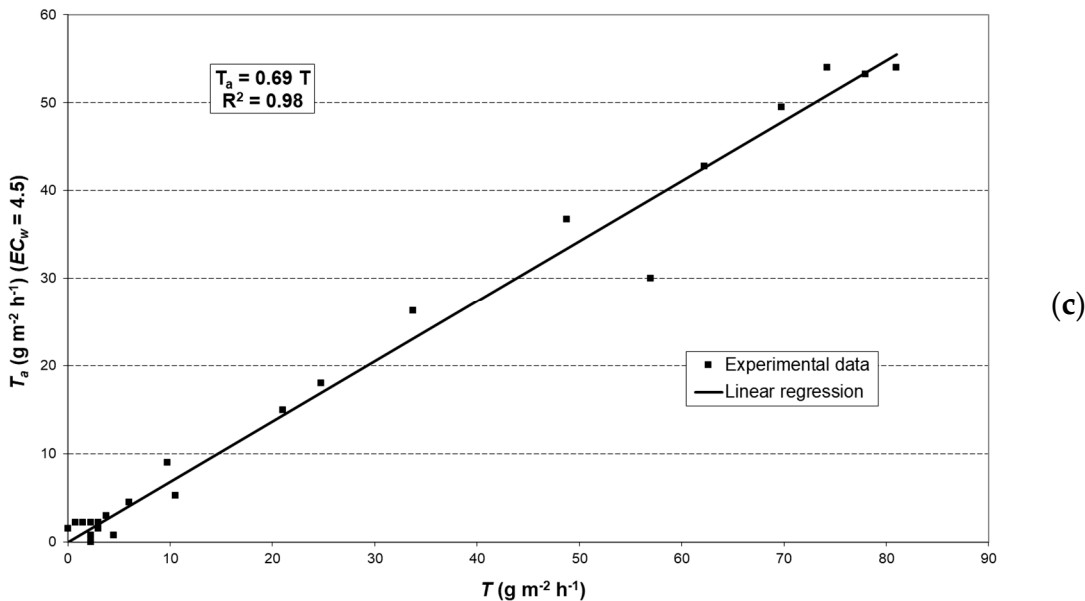

**Figure 4.** Linear regression between the actual transpiration (*Ta*) of the three saline treatments, (**a**) $EC_w$ = 2.5 dS m$^{-1}$, (**b**) $EC_w$ = 3.5 dS m$^{-1}$ and (**c**) $EC_w$ = 4.5 dS m$^{-1}$, and the transpiration of the control treatment (T).

The experimental relationship between the salinity of the nutrient solution and the stress coefficient ($K_s$) was fitted to several models proposed in the literature (linear, convex, and logistic). Both linear and convex models fitted the experimental data well. The linear model performed slightly better ($R^2$ = 0.87 with a *p*-value slightly higher than 0.05) than the convex one ($R^2$ = 0.83). The logistic model, on the contrary, did not have a good performance. This could be because this model was developed for plants significantly differing in water and nutrient demand as well as in the day–night cycle of physiological activity. In addition, in this work, we proposed and tested a new radical $K_s$ curve model. This model overcame the rest since it was able to accurately fit the experimental values with a coefficient of determination ($R^2$) equal to 0.99 and a *p*-value of 0.05. Figure 5 shows the comparison of the models: Linear, convex, and radical. The fitted equations for each model are also shown in the figure.

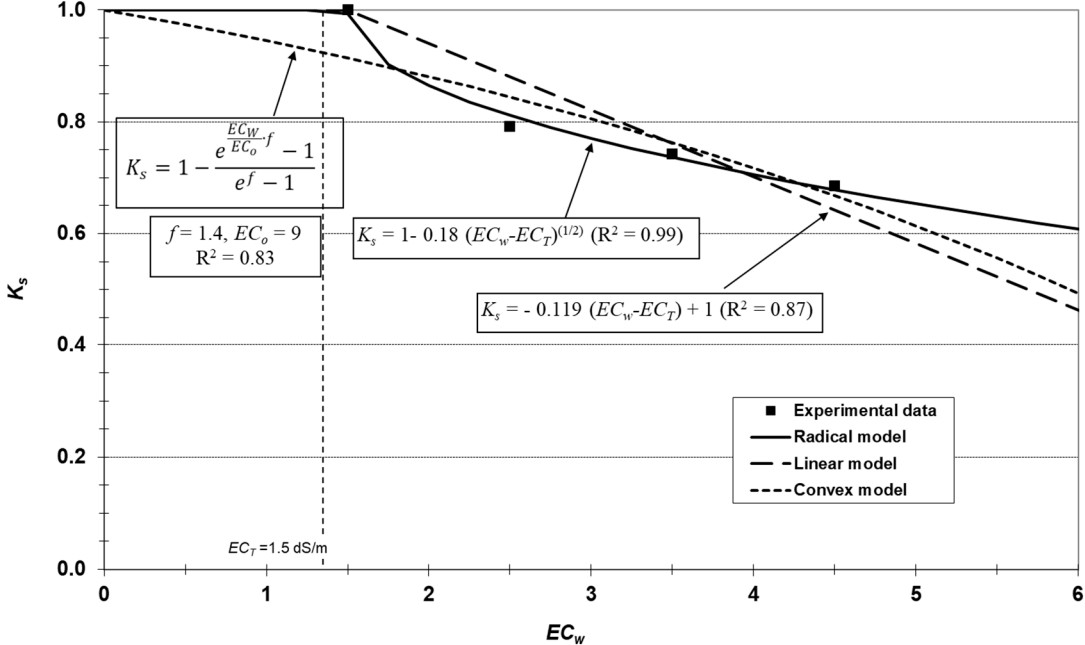

**Figure 5.** Adjusted salinity stress models ($K_s$) by the radical, linear, and convex models.

The proposed equation that relates the salinity stress coefficient, $K_s$, to the electrical conductivity of the nutrient solution for $EC_w$, according to the proposed radical model is given in Equation (5):

$$K_s = 1 - 0.18 \times \sqrt{EC_w - EC_T} . \tag{5}$$

This equation is valid for $EC_w$ values higher than the threshold ($EC_T = 1.5$ dS m$^{-1}$). For $EC_w$ values lower than this threshold value, the $K_s$ values are equal to 1, meaning that no saline stress occurs.

The relative transpiration deficit $(1 - K_s)$ caused a decrease in the growth of the crop. This yield reduction can be expressed in terms of the relative crop yield $(Y_a/Y_M)$. Yield values were measured as aerial dry matter of the Plant The relationship between $(Y_a/Y_M)$ and the relative transpiration deficit $(1 - K_s)$ is shown in Figure 6. The experimental data fitted a two-piece linear relationship well. The experimental relationship showed that for slight transpiration deficits (lower than 18%), no yield reduction occurred. However, for transpiration deficits exceeding this threshold, the relative yield decreased linearly. For severe deficits (higher than 60%), the growth of the plant stopped. The fitted equation is the following:

$$\frac{Y_M}{Y_a} = 1 - 2.34 \times [(1 - K_s) - (1 - K_s)_T]. \tag{6}$$

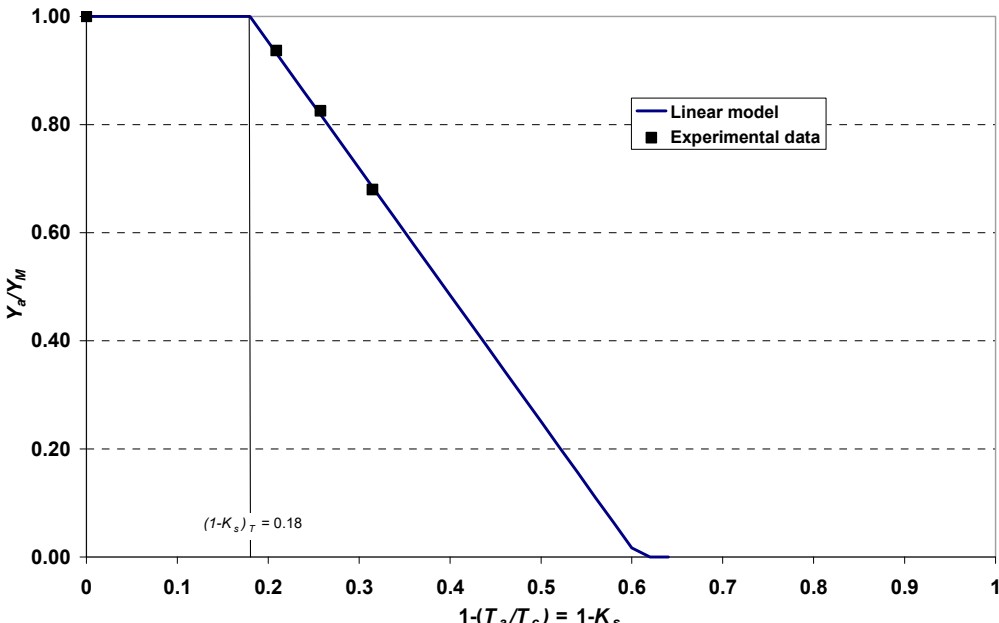

**Figure 6.** Relationship between the relative transpiration deficit $(1 - K_s)$ and the relative crop yield $(Y_a/Y_M)$.

The value of the transpiration deficit threshold $(1 - K_S)_T$ in Equation (6) was equal to 0.18.

A final model that relates the relative crop yield $(Y_a/Y_M)$ and the salinity of the nutrient solution $CE_w$ was obtained by combining Equations (5) and (6). The proposed model is given in Equation (7) and depicted in Figure 7:

$$\frac{Y_a}{Y_M} = 1 - 0.42 \times \left[ \sqrt{EC_W - 1.5} - 1 \right]. \tag{7}$$

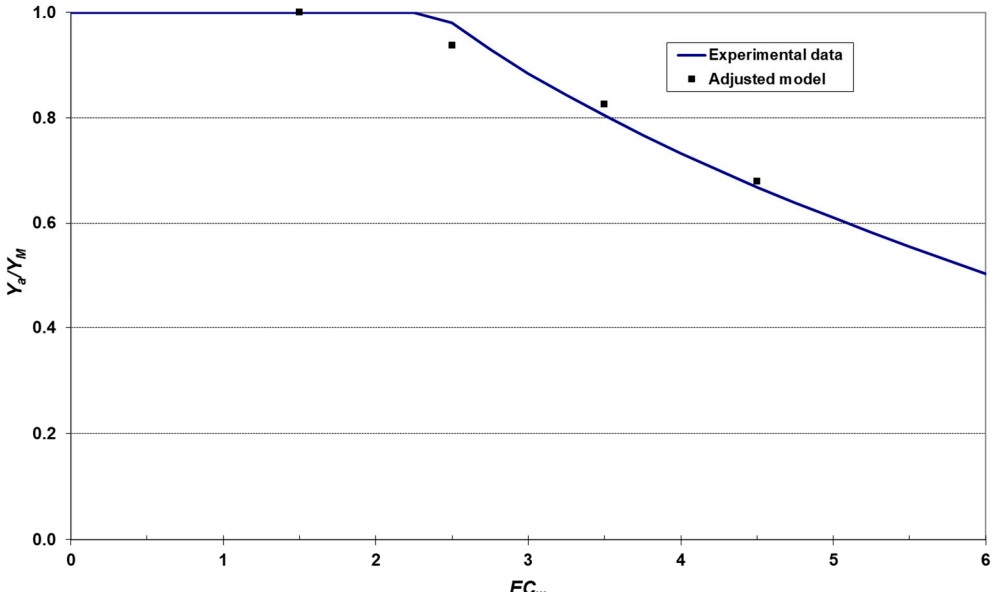

**Figure 7.** Relationship between the electrical conductivity of the nutrient solution ($EC_w$) and the relative crop yield ($Y_a/Y_M$).

## 4. Discussion

The daily water uptake values obtained during the trial showed that the transpiration rate for the control treatment was higher than those obtained for the rest of treatments with higher salinity values. These results coincide with those found in other previous research carried out with different types of crop and climatic conditions. For example, Sharma et al. [43] showed that the transpiration rate of wheat (*Triticum aestivum* L.) grown in pots under natural conditions was reduced with the addition of NaCl. They irrigated with nutrient solutions with electrical conductivities of 4.0, 6.0, and 8.0 dS m$^{-1}$. Also, the unit transpiration rates and relative growth rates of *Populus tomentosa* were restricted by salinity (100 mM NaCl for 8 days and then 200 mM NaCl for 12 days) [44]. However, this depressed transpiration rate can be beneficial because it may prevent plants from undergoing severe water stress, especially in extreme hot and dry environments and high salinity conditions. Some researchers have noticed this effect in a tomato crop [45].

Regarding the evolution of the transpiration over the course of the experiment, in the first week of the experiment (until 7 days after transplant), no significant differences between control and $T_1$ treatments were observed, but there were significant differences with $T_2$ and $T_3$. From the second week on, the plants of all the saline treatments consumed significantly less water than that consumed in the control treatment. Night uptake showed no significant differences except for the second week and for the treatment, $T_0$, in which the night uptake was lower than the rest. This can be considered as a compensation response with respect to the higher day uptake that week. In this sense, Klepper [46] considers that root water replenishment will lag behind transpiration losses, resulting in a slight decrease in plant tissue water content during the daylight period with rehydration occurring during the evening or night as stomata close due to the photoresponse.

As depicted in Figure 1, the results obtained in this experiment for *C. fruticosa* showed the average water uptake during the night was 4.7% 5.9% 6.0%, and 5.8% of the daytime transpiration for $T_0$, $T_1$, $T_2$, and $T_3$, respectively. These average night water uptake values are also consistent with those obtained by other researchers. For example, Van Ieperen [47] concluded that transpiration in tomato plants during the light period was considerably influenced by the osmotic potential applied in the root environment. Furthermore, in a simulation study of nutrient uptake by plants from soilless cultures, Silberbush and Ben-Asher [48] reported instantaneous transpiration losses due to salt accumulation during daytime only. These authors reported an average night water uptake of 0.8 cm$^3$ plant$^{-1}$ night$^{-1}$

for all treatments (so the overall water uptake during the night showed no significant differences between treatments), which meant values between 5% and 6% of the daily water uptake. For tomato and cucumber plants, water uptake during the night (18.00–06.00) was found to be about 12% of that for the whole day [49].

In tomato plants, transpiration during the light period is appreciably influenced by osmotic potential in the root environment levels. In the dark, transpiration decreased after a change in the osmotic potential in the root environment from high (−0.01 MPa) to low (−0.36 MPa), while it increased after the opposite osmotic potential change. However, the differences in transpiration were minimal. At a similar transpiration rate, the plant water deficit was higher in the afternoon than in the morning [47].

The low transpiration values with increasing salinity obtained in this trial are due to the stomatal closure with the involvement of abscisic acid, as Cachorro et al. [50] found in *Phaseolus vulgaris* plants under salt stress. Furthermore, the influence of NaCl (25, 50, 75 mM) on selected physiological responses of kudzu (*Pueraria lobata*) has been evaluated. It was found that photosynthesis and stomatal conductance were significantly reduced as NaCl increased from above 25 mM and that the reduction in stomatal conductance coincided with a similar reduction in transpiration [51]. In wheat (*Triticum aestivum* L. cv. Barkai) plants growing hydroponically, Leidi et al. [52] also found that stomatal conductance decreases with increasing NaCl concentrations in the medium.

Another cause of transpiration reduction is the high *VPD* (−3.56 kPa). 'Daniela' tomato (*Lycopersicon esculentum* Mill.) plants were irrigated with 0 or 50 mM NaCl added to the nutrient solution and grown under natural greenhouse conditions or under applications of fine mist every 8 min during the day. Midday stomatal conductance and net $CO_2$ assimilation rates of salinized misted plants were 3 and 4 times higher, respectively, than those recorded in salinized non-misted plants [53].

Transpiration responses in the short-term are studied to integrate the effects of climate parameters [19,22]. In ornamental plants, there is a wide range of responses of crop transpiration to solar radiation and vapor pressure deficits. In this case, the contribution of radiation (parameter A) is higher than DPV (parameter B). It is important to highlight the immediate response of transpiration in relation to the modification of radiation, between 10 and 13 h (Figure 2).

Katsoulas et al. [23] stated that in order to accurately predict short-term variations of a rose canopy's transpiration rate and conductance from greenhouse environmental conditions, the magnitude and diurnal variation of *VPD* during the day must be accounted for. As a first approximation, and regarding the dynamics of transpiration, this effect may be due to a stomatal conductance decrease [54]. The stomatal conductance decrease could explain, by itself, the diminution of transpiration related to the cellular solution's colligative properties (lower vapor pressure under higher salinity) of the dissolution in the cells. For this reason, following Raoult's law, it requires more energy (solar energy) per unit of transpired water. Stomatal conductance of pine trees is more strongly affected by vapor pressure deficit than radiation, under salt treatment. For example, *Phillyrea latifolia* has markedly lower concentrations of $K^+$ and soluble carbohydrates than control leaves under salinity conditions [55]. As drought increases, osmotic potential at full turgor decreases and the total solute concentrations increase in leaves, indicating osmotic adjustment. Decreases in leaf starch concentrations and concomitant increases in hexose sugars and sucrose suggest a shift in carbon partitioning in favor of soluble carbohydrates [56]. Variations of leaf water potential are closely correlated with changes in malate, mannitol, and the concentration of the well-known osmoticum $K^+$ [57], which can correspond to type II solutes' behavior and physicochemical properties. Nativ et al. [58] found that stomatal conductance is negatively correlated with air *VPD* under drought conditions, whereas under irrigation in the field, it was correlated with solar radiation. Similar results were found in $T_3$ and $T_4$, but the reason was salinity in this case.

Also, from 16.00 h, all treatments presented a similar response. The transpiration of treatments "cherry"-type tomato plants, which were well watered (irrigated with freshwater, EC 0.7 dS m$^{-1}$) and put under moderate saline stress (2.5 dS m$^{-1}$), increased until the warmest hours of the day, and

decreased thereafter. Treatments with higher saline stress showed lower values of transpiration and a lower time-difference. Differences between treatments decreased throughout the day [59].

A radical function was proposed in this work to describe the relationship between yield and salinity. The developed model was based on a separate consideration of the two main processes involved in the yield reduction, i.e., the effect of the salinity of the nutrient solution on the transpiration reduction and the effect of the transpiration reduction on the crop yield. This approach was proven to be appropriate since the proposed model fitted the experimental data well. The relationship between the relative crop yield and the conductivity of the nutrient solution was shown to not be linear, in the same way as Correia et al. [41] had previously observed. Different salinity stress curves proposed in the literature (i.e., linear, convex, and logistic) were tested in this research. However, a new radical stress model was shown to be best suited to adjusting the experimental results.

## 5. Conclusions

The final conclusions withdrawn from this work are the following: Day transpiration constitutes around 95% of the total water uptake by plants. The transpiration of *C. fruticosa* grown in a container under no salinity stress was modelled by using the Penman–Monteith equation modified by Baille et al. (1994). The increment of the salt concentration in the nutrient solution caused a diminution of the day transpiration of *C. fruticosa* "Red Edge" plants. The effect of the nutrient solution salinity was taken into account by applying a salinity stress coefficient ($Ks$). Different salinity curves proposed in the literature were tested and fitted to the experimental data. Linear and convex shapes performed satisfactorily, nevertheless, the root-square $Ks$ model proposed in this work was found to be the best suited to account for the reduction in transpiration due to salinity stress. This model could be a useful tool to improve crop irrigation management.

The model presented in this work provides a novel procedure to perform an estimation of the transpiration of the crop as a function of the salinity of the nutrient solution and its effect on yield reduction. However, more experimental research must be done to calibrate the model under different environmental and management conditions and for other types of potted plants or substrate culture in order to estimate the effect of salinity on the transpiration of the crop and on its yield reduction. The results of this work are useful to improve the sustainability of the system through improvements of the accuracy of water consumption estimation and the reduction of the water irrigation supply to the plant, thus decreasing the environmental impact of leachates.

**Author Contributions:** Investigation, data curation B.M.P.; formal analysis, data curation, writing—original draft preparation, J.R.; formal analysis, validation, methodology, J.M.; conceptualization, writing—review and editing, project administration, M.T.L.; Experiment design, development of transpiration model, supervision of the writing and coordination of work, F.A.

**Funding:** This research was funded by the European Union LIFE program under the project DESEACROP LIFE INV/ES/000341 and the Spanish Ministry for Science, Innovation and Universities program RETOS INVESTIGACIÓN under the project RIDESOST: SOSTenibilidad agro-fisiológica, ambiental y económica del RIego con agua marina DEsalinizada en cítricos y sistemas hidropónicos semicerrados en cultivos de invernadero (AGL2017-85857-C2-1-R).

**Acknowledgments:** We want to acknowledge the supporters of this research.

**Conflicts of Interest:** The authors declare no conflict of interest.

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
