# Peer review of "Sustainable Irrigation Management of Ornamental Cordyline Fruticosa “Red Edge” Plants with Saline Water"

_sustainability, doi:10.3390/su11133751_

Round 1

Reviewer 1 Report

see attached .pdf file

Comments

You have analyzed water use and growth response of Cordyline fruticosa plants irrigated with nutrient solution differing in NaCl content. As a result you are presenting a mathematical model allowing to calculate the observed relationship between plant growth and use of irrigation water. A special aspect is that you have performed the experiments in a greenhouse located in the Mediterranean region. Growth conditions significantly differ from controlled conditions in use in the norther part of Europe, for instance. Thus, your results are special and in the presented form exclusively apply to your experimental plant species and the conditions in your greenhouse.
You are discussing different mathematical models used by other research teams. Again, these models apply only for the experimental conditions in use and for plants having similar physiological attributes. You have correctly found out that models do not match, if they have been developed for plants significantly differing in water and nutrient demand as well as in day-night cycle of physiological activity. You might have explained this reason more clearly. Your result is important for horticulturists interested in growing Cordyline fruticosa in your area or in a similar climate. In this context it is not important to discuss details such as hormonal control of photosynthesis and respiration. The most important aspect is that you have proven that predictions made by your model are reliable within the concentration range of NaCl you have tested.
Please check your discussion. It is quite wordy in parts. The same holds true for the introduction chapter.
You have used an ornamental plant that can be eaten. Nevertheless I wonder, whether the word „crop“ should be used.
The size and number of leaves is most important when selling your ornamental plant. Therefore you have measured leaf area as a parameter. But the term „yield“ preferentially refers to biomass, fresh weight, etc. - Why don’t you discuss production of leaf area, i.e. the parameter you are interested in?

Author Response

The answers to your comments are in the attached document

Reviewer 2 Report

I have two major concerns regarding this paper. The paper is trying to evaluate effect of salinity on transpiration and yield. Also whether night transpiration is more or daytime.

I think all these aspects are well studied and established that due to high osmotic pressure, root have difficulty in extracting water from soil under saline conditions. Therefore transpiration and ultimately crop yield is reduced. Similarly, day time transpiration is proven to be higher than the night time. Therefore, authors should clearly define why it is necessary to do this research again. Introduction of the paper focuses more on theory behind these processes rather than the need.

The results show mainly non-significant results and the difference in transpiration rates with increasing salinity levels are very low. T2 and T3 are showing almost same results.

In the conclusions, authors should clearly define their recommendations regarding irrigation with saline water as they claim that this study is useful to improve crop irrigation management. Lines 419-421 seems incomplete sentence. Needs completion.

It would be good if authors can also explain how the results of this study are related to the aspects of sustainability 

Some specific points

Line 41: Add FAO reference.

Introduction is bit lengthy and can be shortened by avoiding theoretical discussion.

Line 160: How much NaCl was added to get each salinity level. This information is needed.

Line 163: Table 1. It should be Electrical conductivity.

Table 1: E.C. should be replaced by EC.

Line 185: Not clear. Why measurements were done after the trial for measuring hourly transpiration.

Units need to be consistent. They should all be dS m-1 and not dSm-1

Line 210: It should be saturated -paste extract.

Figure 1: Explain mL/plant/day??

All Figures need to be consistent in titles an legends.

Figure 3: On hour 10, explain the extra lower transpiration values.

Spellings need to be checked carefully. 

Units need to be consistent. somewhere it is written cm2 and others as cm2

Author Response

(The authors gave the same response as above.)

Reviewer 3 Report

The manuscript sustainability-515061  entitled “Sustainable irrigation management of ornamental Cordyline fruticosa “Red Edge” plants with saline water” by Plaza et al., deals with an interesting and quite important subject regarding the irrigation of pot plants under greenhouse condition with saline irrigation water. In particular, the authors focus their experiment on two different aspects: a) evaluate the effect of irrigation with saline water on water consumption, transpiration and productivity of Cordyline fruticosa var. “Red Edge”; b) to model the reduction of productivity of this plant as the salinity of the nutrient solution increases.

The objectives of the study are interesting because the problem of irrigation saline water is important in the Mediterranean areas, especially in greenhouse production district, and its importance will increase more and more in the future due to climate change.

The manuscript should be greatly improved in order to be published. For this reason, I suggest to publish the paper after it has been made the major revisions with suggested changes.

First of all, I suggest the authors to carefully read and revise the language of the manuscript. In particular, I suggest to reduce the length of some sentences in order to increase their comprehension.

The introduction its good enough though in some passages too generic. I suggest to reduce and rewrite some parts in order to be more focused on the topic. The objective of the study clearly presented. Abstract clearly present the activities and the principal results.

Keywords are appropriate.

Material and methods should be improved and some experimental aspects should be clarified. Some parts that are too descriptive should be removed or moved to the introduction. Below are some specific comments.

The results are sufficiently detailed.

Discussion although it presents comparisons with the results of other studies it lacks of a clear explanation of what was observed and doesn’t is organized as a single thread. Moreover, the modelling of data was not discussed. Therefore, I suggest deepening this section.

Below are my specific comments:

Line 32: approximate the area invested especially because the reference that is reported is a little old.

Line 35-37: They can be combined into one sentence.

Line 45: change “some” with “several”.

Line 45: change “way of improving” with “way to improve”.

Line 62-65: Add a phrase about the leaching requirement.

Line 110-114: Should be deleted.

Line 151-154: Please explain better.

Line 160: ECw  “w” subscript throughout the manuscript.

Line 161: correct the chlorine ion.

Line 178: explicit what did you weighted.

Line 185-186: How did you registered the weight during the night?

Lines 190-191: delete “(range 30 to 190 300 ºC)”.

Line 197 and forward: correct the units of measurement by adding the spaces between one unit and other.

Lines 210-221: Please rewrite describing only the method did you applied.

Lines 226-238: Too descriptive. Please leave only the part where did you describe the applied approach.

Line 249: Table 2; Please remove LSD and P values or on the contrary explain above in the text where statistical analysis was applied.

Line 252: add significance level.

Lines 253-254: Please, add % differences among treatments inside brackets.

Line 254: The level of significance level is wrong. Should be > because not significant.

Lines 254-255: Should be divided in two phrases.

Line 256: “This” should be in lower case.

Lines257-258: Should be moved at the beginning of the paragraph.

Lines 259: The resolution of the graph is poor. Moreover, I suggest to present this data as a bar chart.

Line 260: delete “daily”.

Line 266: change the first “and” with a comma.

Line 267: change “ to were “ with “ were”.

Line 269: put the level of significance inside brackets.

Lines 289-291: Please, simplify this phase and explicit the models applied.

Line 292: change “greater” with “higher”.

Line 293: check “performance”.

Lines 296-297: please, delete.

Line 309: delete “found in this work”

Lines 319-320: please delete.

Lines 325-332: Please, rewrite this paragraph more focused on your experiments, or alternatively delete.

Line 333: change “show” with “showed”.

Line 335: delete “work”.

Lines 340-341: Connect this phase to the previous.

Line 345: remove the comma.

Line 359: 20px3, 3 should be apex.

Line 382: check CO2.

Lines 393-394: Please, rewrite.

Lines 398-340: Please link it better to your research or delete.

Line 400: do you mean increase? Change “develops” with “increase”.

Line 407: delete “in Acacia saligna” If not fundamental.

Lines 408-409: Please explain better or rewrite.

Line 421: Please add the year of publication of the model.

Line 423: scientific names should be in italics

Author Response

(The authors gave the same response as above.)

Round 2

Reviewer 2 Report

Still introduction is very long

Lines 204-210 are not clear. In line 205, it should be quantify or quantity? please check

Lines 215-220. why they use different approach than generally accepted approach? Needs strong justification.

In Figure 1, what is mL?

The explanation given by authors about the novelty of this work is not very strong. As they agreed with my earlier comments, this work is well established for many regions under different soil and water conditions.We need better explanation why to repeat this work.

Authors also agree that the results are non-significant so what is the rational to do this work?  

Author Response

Reply to reviewer

All the suggestions made by the reviewer were considered in the preparation of this revised version.

Reviewer 2

Still introduction is very long

Lines 204-210 are not clear. In line 205, it should be quantify or quantity? please check

Corrected

Lines 215-220. why they use different approach than generally accepted approach? Needs strong justification.

In Figure 1, what is mL?

Millilitres of water (Legend)

The explanation given by authors about the novelty of this work is not very strong. As they agreed with my earlier comments, this work is well established for many regions under different soil and water conditions. We need better explanation why to repeat this work.

Either we were not clear enough in our previous response or the reviewer has misunderstood our explanations. We really agreed about the fact that the basic concepts involved in the effect of the salinity on the crop transpiration and yield are known. However, we believe that there is a strong gap in the development of models that effectively quantify this effect for different cropping conditions and make rational recommendations about the proper irrigation management with these saline waters. The use of general recommendations, obtained for other cropping conditions, to manage substrate culture in Mediterranean greenhouse systems, usually leads to apply excessive amounts of irrigation water that may be detrimental not only for the crop but also to the environment. Our work aims to fill this existing gap of reliable modelling of the salinity management in such conditions. No way we are repeating other experimental works. The objective, experimental set-up, applied methods and developed models are completely original and are of great interest for the management of saline irrigation water in the above-mentioned conditions.

Authors also agree that the results are non-significant so what is the rational to do this work?  

Again, our response was not as clear as it should be. We did not mean at all that our results were non-significant. What we really meant was that our experimental work is limited to the conditions in which the experiment was conducted. However, this is so for most research works carried out in agriculture. Nevertheless, the results and conclusions along with the developed model can be directly applied to other regions with similar climate conditions (Mediterranean climate) and other similar cropping systems. In addition, and with a proper calibration, the model developed in our research could be extended to other different climate and cropping conditions. One of the main advantage of the proposed model is that it decomposes the effect of salinity on yield on two separate processes, i.e. the effect of salinity on the crop transpiration and the effect of the transpiration reduction on the crop yield, in such a way that the comprehension of the general process is deeper and the generalization of the model to other conditions is thus more straightforward.
